# SEMI-SUPERVISED FEW-SHOT LEARNING WITH PROTOTYPICAL RANDOM WALKS

## ABSTRACT

Learning from a few examples is a key characteristic of human intelligence that inspired machine learning researchers to build data efficient AI models. Recent progress has shown that few-shot learning can be improved with access to unlabelled data, known as semi-supervised few-shot learning(SS-FSL). We introduce an SS-FSL approach, dubbed as Prototypical Random Walk Networks(PRWN), built on top of Prototypical Networks (PN) (Ren et al., 2018). We develop a random walk semi-supervised loss that enables the network to learn representations that are compact and well-separated. Our work is related to the very recent development on graph-based approaches for few-shot learning. However, we show that compact and well-separated class representations can be achieved by modeling our prototypical random walk notion without needing additional graph-NN parameters or requiring a transductive setting where collective test set is provided (e.g., Kim et al. (2019)). Our model outperforms prior art in most benchmarks with significant improvements in some cases. For example, in a mini-Imagenet 5-shot classification task, we obtain 69.65% accuracy to the 64.59% state-of-the-art. Our model, trained with 40% of the data as labelled, compares competitively against fully supervised prototypical networks, trained on 100% of the labels, even outperforming it in the 1-shot mini-Imagenet case with 50.89% to 49.4% accuracy. We also show that our model is resistant to distractors, unlabeled data that does not belong to any of the training classes, and hence reflecting robustness to labelled/unlabelled class distribution mismatch. We also performed a challenging discriminative power test, showing a relative improvement on top of the baseline of $\approx$14% on 20 classes on mini-Imagenet and $\approx$60% on 800 classes on Omniglot. Our code will be released upon acceptance.

## 1 INTRODUCTION

Humans are capable of learning complex skills efficiently and quickly, raising a challenging question to scientists and philosophers: "How do our minds get much from so little?" (B Tenenbaum et al., 2011). In contrast, artificial learners require large amounts of labeled data to reach comprable levels (Dodge & Karam, 2017). This gap motivated progress in few-shot learning (FSL) and semi-supervised learning (SSL). Semi-supervised learning develops techniques that benefit from an abundance unlabeled of data for training. It has gained a big interest in the 90s and the early 2000s, guided not only with applications in natural language problems and text classification (Yarowsky, 1995; Nigam & Kosovichev, 1998; Blum & Mitchell, 1998; Collins & Singer, 1999; Joachims, 1999), but also in computer vision as in segmentation Shi & Malik (2000); Li et al. (2004). Few-shot learning is an artificial learning skill of rapidly generalizing from limited supervisory data (few labeled examples), typically without the using of any unlabeled data (Koch et al., 2015; Miller et al., 2000; Lake et al., 2011). Our work is at the intersection between few-shot learning and semi-supervised learning where we augment the capability of few-shot artificial learners with a learning signal derived from unlabeled data.

*Semi-supervised Few-shot Learning (SS-FSL):* Few-shot learning methods typically adopt a supervised learning setup (e.g., (Vinyals et al., 2016; Ravi & Larochelle, 2017b; Snell et al., 2017)), very recently, Ren et al. (2018) and Zhang et al. (2018) developed Semi-supervised few-shot learning approaches that can leverage additional unlabeled data. The machinery of both approaches adopts a meta-learning episodic training procedure with integrated learning signals from unlabeled data. Ren

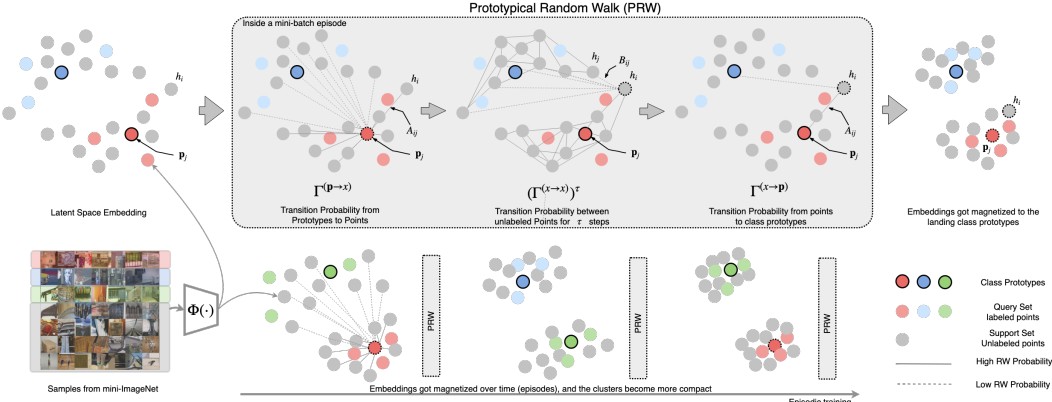

Figure 1: Our PRW aims at maximizing the probability of a random walk begins at the class prototype $\mathbf{p}_j$, taking $\tau$ steps among the unlabeled data, before it lands to the same class prototype. This results in a more discriminative representation, where the embedding of the unlabeled data of a particular class got magnetized to its corresponding class prototype, denoted as *prototypical magnetization.*

et al. (2018) build on top of prototypical networks(PN) (Snell et al., 2017) so better class prototypes can be learned with the help of the unlabeled data. Zhang et al. (2018) proposed a GAN-based approach, Meta-GAN, that helps making it easier for FSL models to learn better decision boundaries between different classes.

In this work, we propose Prototypical Random Walks (PRW), as an effective graph-based learning signal derived from unlabelled data. Our approach improves few-shot learning models by a random walk through the embeddings of unlabeled data starting from each class prototype passing through unlabeled data in the embedding space and encourages returning to the same prototype at the end of the prototypical walk (*cf.* Fig. 1). This PRW learning signal promotes a latent space where points of the same class are compactly clustered around their prototype, while being well isolated from other prototypes. We sometimes refer to this discriminative attraction to class prototypes as *prototypical magnetization.*

Since the PRW loss is computed over a similarity graph involving all the prototypes and unlabelled points in the episode, it takes a global view of the data manifold. Due to the promoted *prototypical magnetization* property, this global view enables more efficient learning of discriminative embeddings from few examples, which is the key challenge in few-shot learning. In contrast, there are local SSL losses, where the loss is defined over each point individually, most notable of those approaches is the state-of-the-art Virtual Adversarial Training (VAT) (Miyato et al., 2018). We show that in the FSL setting, our global consistency guided by our prototypical random walk loss adds a learning value compared to local consistency losses as in VAT (Miyato et al., 2018).

**Contribution.** We propose Prototypical Random Walk Networks (PRWN) where we promote *prototypical magnetization* of the learning representation. We demonstrate the effectiveness of PRWN on popular few-shot image classification benchmarks. We also show that our model trained with a fraction of the labels is competitive with PN trained with all the labels. Moreover, we demonstrate that our loss is robust to "distractor" points which could accompany the unlabeled data yet not belong to of any of the training classes of the episode.

## 2 APPROACH

We build our approach on top of Prototypical Networks (PN) (Snell et al., 2017) and augment it with a novel random walk loss leveraging the unlabelled data during the *meta-training* phase. The key message of our work is that more discriminative few-shot representations can be learned through training with prototypical random walks. We maximize the probability of a random walk which starts from a class prototypes and walk through the embeddings of the unlabeled points to land back to the same prototype; see Fig. 1. Our random walk loss enforces the global consistency where the

overall structure of the manifold is considered. In this section, we detail the problem definition and our loss.

## 2.1 PROBLEM SET-UP

The few-shot learning problem may be formulated as training over a distribution of classification tasks $\mathcal{P}_{train}(\mathcal{T})$, in order to generalize to a related distribution of tasks $\mathcal{P}_{test}(\mathcal{T})$ at test time. This setting entails two levels of learning; *meta-training* is learning the shared model parameters(meta-parameters) to be used on future tasks, *adaptation* is the learning done within each task. Meta-training can be seen as the outer training loop, while adaptation being the inner loop.

Concretely, for $N_s$-shot $N_c$-way FSL, each task is an episode with a support set $\mathcal{S}$ containing $N_s$ labelled examples from each of $N_c$ classes, and a query set $\mathcal{Q}$ of points to be classified into the $N_c$ episode classes. The support set is used for adaptation, then the query set is used to evaluate our performance on the task and compute a loss for meta-training.

To run a standard FSL experiment, we split our datasets such that each class is present exclusively in one of our train/val/test splits. To generate a training episode, we sample $N_c$ training classes from the train split, and sample $N_s$ samples from each class for the support set. Then we sample $N_q$ images from the same classes for the query set. Validation and test episodes are sampled analogously from their respective splits.

Following the SS-FSL setup in Ren et al. (2018); Zhang et al. (2018); Liu et al. (2019), we split our training dataset into labelled/unlabelled; let $\mathcal{D}_{k,L}$ denote all labeled points $x \in class(k)$, and $\mathcal{D}_{k,U}$ be all unlabeled points $x \in class(k)$. Analogous notation holds for our support and query set, $\mathcal{S}$ and $\mathcal{Q}$. To set up an semi-supervised episode, we simply need to add some unlabelled data to the support set. For every class $c$ sampled for the episode, we sample $N_u$ samples from $\mathcal{D}_{c,U}$ and add them to $\mathcal{S}$. In order to make the setting more realistic and challenging, we also test our model with the addition of *distractor* data. Those are unlabelled points added to the support set, but not belonging to the episode classes. We simply sample $N_d$ additional classes, and sample $N_u$ points from each class to add to the support set. We present pseudo-code for episode construction in appendix B.

It is worth mentioning that the unlabelled data may be present at either train or test time, or both. At training time, we want to use the unlabelled data for meta-training i.e. learning better model parameters. For unlabelled data at test time, we want to use it for better adaptation, i.e. performing better classification on the episode's query set. Our loss operates on the meta-training level, to leverage unlabelled data for learning better meta-parameters. However, we also present a version of our model capable of using unlabelled data for adaptation, by using the semi-supervised inference from (Ren et al., 2018) with our trained models.

**Prototypical Networks.** Prototypical networks (Snell et al., 2017) aim to train a neural network as an embedding function mapping from input space to a latent space where points of the same class tend to cluster. The embedding function $\Phi(\cdot)$ is used to compute a *prototype* for each class, by averaging the embeddings of all points in the support belonging to that class, $\mathbf{p}_c = \frac{1}{|\mathcal{S}_{c,L}|} \sum_{x_i \in \mathcal{S}_{c,L}} \Phi(x_i; \theta),$

where $\mathbf{p}_c$ is the prototype for our $c$-th class, and $\theta$ represents our meta-parameters. Once prototypes of all classes are obtained, query points are also embedded to the same space, and then classified based on their distances to the prototypes, via a softmax function. For a point $x_i$, with an embedding $h_i = \Phi(x_i; \theta)$, the probability of belonging to class $c$ is computed by

$$z_{i,c} = p(y_c|x_i) = \frac{\exp\left(-d(h_i, \mathbf{p}_c)\right)}{\sum_{j=1}^{N_c} \exp\left(-d(h_i, \mathbf{p}_j)\right)}, \quad \tilde{\mathbf{p}}_c = \frac{\sum_{x_i \in S_U \cup S_L} h_i \cdot z_{i,c}}{\sum_{i=1}^{N} z_{i,c}}. \tag{1}$$

where $d(\cdot, \cdot)$ is the Euclidean distance. In the semi-supervised variant (Ren et al., 2018), PN use the unlabelled data to *refine* the class prototypes. This is achieved via a soft K-means step. First, the class probabilities for the unlabelled data $z_{i,c}$ are computed as in Eq.1, and the labelled points have a hard assignment, i.e. $z_{i,c}$ is 1 if $x_i \in class(c)$ and 0 otherwise. Then the updated prototype $\tilde{\mathbf{p}}_c$ is computed as the weighted average of the points assigned to it; see Eq. 1. We can see this as a task *adaptation* step, which does not directly propagate any learning signal from the unlabelled points to our model parameters $\theta$. In fact, it might be used only at the inference time, and results from (Ren et al., 2018) show that it provides a significant improvement when used as such. When used during

*meta-training* by updating the model parameters from the unlabelled data, the performance improves only marginally (i.e., from 49.98% to 50.09% on mini-imagenet (Vinyals et al., 2016)). While this approach is powerful as the *adaptation* step, it fails to fully exploit the unlabelled data during *meta-training*. *SS-FSL with adaption at test time.* Our approach also allows using the former K-means refinement step at inference time, analogous to the 'Semi-supervised inference' model from (Ren et al., 2018). Orthogonal to (Ren et al., 2018), our approach can be thought as a meta-training regularizer that brings discriminative global characteristics to the learning representation efficiently .

## 2.2 PROTOTYPICAL RANDOM WALK

Given the class prototypes $\mathbf{p}_c$, computed using the labeled data in the support set $\mathcal{S}_L$, and the embeddings $h_i$ of unlabeled support set $\mathcal{S}_U$, we construct a similarity graph between the unlabeled points' embeddings and the prototypes. Our goal is to have points of the same class form a compact cluster in latent space, well separated from other classes. Our Prototypical Random Walk(PRW) loss aims to aid this by efficiently magnetizing the unlabelled embeddings around the class prototypes(*cf.* Fig. 1).

This notion is translated into the idea that a random walker over the similarity graph rarely crosses class decision boundaries. Here, we do not know the labels for our points or the right decision boundaries, so we cannot optimize for this directly. Inspired by (Haeusser et al., 2017), we basically imagine our walker starting at a prototype, taking a step to an unlabeled point, and then stepping back to a prototype. The objective is to increase the probability that the walker returns to the same prototype it started from; we will refer to this probability as the *landing probability*. Additionally, we let our walker taking some steps between the unlabelled points, before taking a step back to a prototype.

Concretely, for an episode with $N$ class prototypes, and $M$ unlabeled points overall, let $A \in \mathbb{R}^{M \times N}$ be the similarity matrix, such that each row contains the negative Euclidean distances between the embedding of an unlabelled point and the class prototypes. Similarly, we compute the similarity matrix between the unlabeled points $B \in \mathbb{R}^{M \times M}$. Mathematically speaking, $A_{i,j} = -\|h_i - \mathbf{p}_j\|^2, B_{i,j} = -\|h_i - h_j\|^2$ where $h_i = \Phi(x_i)$ is the embedding of the $i$-th unlabeled sample, and $\mathbf{p}_j$ is the $j$-th class prototype. The diagonal entries $B_{i,i}$ are set to a small enough number to avoid self-loop.

Transition probability matrices for our random walker are calculated by taking a softmax over the rows of similarity matrices. For instance, the transition matrix from prototypes to points is obtained by softmaxing $A^T$, $\Gamma^{(\mathbf{p} \to x)} = \text{softmax}(A^T)$, such that $p(x_i|\mathbf{p}_j) = \Gamma_{i,j}^{(\mathbf{p} \to x)}$. Similarly, transition from points to prototypes $\Gamma^{(x \to \mathbf{p})}$, and transitions between points $\Gamma^{(x \to x)}$, are computed by softmaxing $A$, and $B$, respectively. Now, we define our random walker matrix as

$$T^{(\tau)} = \Gamma^{(\mathbf{p} \to x)} \cdot (\Gamma^{(x \to x)})^\tau \cdot \Gamma^{(x \to \mathbf{p})}, \tag{2}$$

where $\tau$ denotes the number of steps taken between the unlabelled points, before stepping back to a prototype. An entry $T_{i,j}$ denotes the probability of ending a walk at prototype $j$ *given* that we have started at prototype $i$, and the $j$-th row is the probability distribution over ending prototypes, given that we started at prototype $j$. The diagonal entries of $T$ denote the probabilities of returning to the starting prototype; our landing probabilities. Our goal is to maximize those by minimizing a cross-entropy loss between the identity matrix $I$ and our random walker matrix $T$, dubbed as $L_{walker}$[1]

$$\mathcal{L}_{walker} = \sum_{i=0}^{\tau} \alpha^i \cdot H(I, T^{(i)}), \quad \mathcal{L}_{visit} = H(\mathcal{U}, P), \quad \mathcal{L}_{RW} = \mathcal{L}_{walker} + \mathcal{L}_{visit}, \tag{3}$$

where $H(I, T) = -\frac{1}{N_c} \sum_{i=0}^{N_c} \log T_{i,i}$, and $\alpha$ is an exponential decay hyperparameter. However, one issue with $L_{walker}$ loss, is that we could end up visiting a small subset of the unlabelled points. To remedy this problem, (Haeusser et al., 2017) introduce a 'visit loss', pressuring the walker to visit a large set of unlabeled points. Hence, we assume that our walker is equally likely to start at any prototype, then we compute the overall probability that each point would be visited when we

---

[1]To be exact, this is the average cross-entropy between the individual rows of $I$ and $T$, since those are probability distributions.

step from prototypes to points $P = \frac{1}{N_c} \sum_{i=0}^{N_c} \Gamma_i^{(\mathbf{p} \to x)}$, where $\Gamma_i^{(\mathbf{p} \to x)}$ represents a column of the matrix. Then we add $\mathcal{L}_{visit}$ as the standard cross-entropy between this probability distribution and the uniform distribution $\mathcal{U}$. Hence, our final random walk loss is $\mathcal{L}_{RW}$ is the sum of $\mathcal{L}_{walker}$ and $\mathcal{L}_{visit}$; see Eq 3.

**Overall Loss.** To put it all together, our objective function can be written as $\arg\min_\theta \mathcal{L}_S + \lambda \mathcal{L}_{RW}$, where $\lambda$ is a regularization parameter. While gradient of $\mathcal{L}_S = -\sum_{i=0}^{Q_L} y_i \log z_{i,c}$ provides the supervised signal, the gradient of $\mathcal{L}_{RW}$ encourages the *"prototypical magnetization"* property guided by our random walk. This loss is minimized in expectation over randomly sampled semi-supervised episodes from our training data.

## 3 RELATED WORKS

**PRWN Inspiration from Literature.** Recently, Ren et al. (2018) introduced the SS-FSL setting, and augmented PN with an adaptation step which leverages the unlabelled data. Later, Zhang et al. (2018) introduced another SS-FSL approach, dubbed MetaGAN, built on top of relation networks (Sung et al., 2018), but employs an additional generator network (GANs) (Goodfellow et al., 2014). Analogous to (Dai et al., 2017), the fundamental idea is to use the GAN generator to generate points in low density areas around the data. The discriminator is required to classify points into the $K$ classes plus detect the fake points. This setup induces the discriminator to place decision boundaries in low density areas. On the other hand, SSL contains a rich toolbox of discriminative principles/techniques which are simple, effective, and well-studied, yet have never been applied to the FSL setting (Miyato et al., 2018; Kamnitsas et al., 2018; Haeusser et al., 2017). We draw inspiration from those techniques, and design a state-of-the-art model which does not require an additional generator or adversarial training.

**Local consistency.** Unlabelled data gives information about the data density $P(x)$, and we somehow wish to use that information to improve on a classifier modeling $P(y|x)$. Discrminative methods commonly do this by assuming some relation between the label distribution and the data density, and designing a loss to enforce this prior. Most methods can be roughly said to enforce either *local consistency* or *global consistency* (Chapelle et al., 2010; Zhou et al., 2004). Local consistency means points which are close together hold similar labels. Global consistency assumes that point on the same structure have the same label. Much of local consistency methods are perturbation-based; where the objective is to minimize $D(f_\theta(x), f_\theta(\tilde{x}))$, where $D$ is a distance function, $f$ the classifier, and $\tilde{x}$ is a perturbed version of $x$. VAT(Miyato et al., 2018) is one such method, where the perturbation made to $x$ is an adversarial one.

**Global Consistency and Graph-based Methods.** Our loss falls into the global consistency methods, more specifically graph-based methods (Zhu et al., 2005; Zhou et al., 2004; Kamnitsas et al., 2018; Haeusser et al., 2017). These methods operate over a graph with an adjacency matrix $W$, where $W_{i,j}$ is the similarity between samples $x_i, x_j \in \mathcal{D}_L \cup \mathcal{D}_U$. This graph is meant to reveal the structure of the data manifold, then points close over the manifold (high $W_{i,j}$), should have similar labels. Kamnitsas et al. (2018) and Haeusser et al. (2017) both use a random walk formulation, and encourage some notion of consistent walks, however, class prototypes are not involved. In Haeusser et al. (2017) the walker starting at a labelled point of class $c$ is required to return to any point of class $c$. In (Kamnitsas et al., 2018) improve on this approach by first performing a label propagation to get labels for the unlabelled points, then an *ideal* transition matrix is constructed using those labels, and finally the cross-entropy between this ideal walker transitions, and the actual walker transitions is minimized. By involving prototypes in the graph, as the representatives of classes, our procedure is simple, and better suited for prototypical networks; as we are ultimately interested in magnetizing the points around the prototypes.

**Transductive/Semi-supervised Adaptation Approaches.** Apart from MetaGAN (Sung et al., 2018), existing semi-supervised/transductive methods, use the unlabelled/query data as part of the classification procedure e.g., by defining better semi-supervised prototypes (Ren et al., 2018) or by replacing the K-means step with label propagation (Douze et al., 2018; Liu et al., 2019). None of those methods derive an *additional* training signal from the unlabelled data. In other words, our method focuses on **meta-training**, the prior methods focus on **adaptation** procedures which can leverage additional data. Kim et al. (2019) recently proposed an edge-labeling graph neural network (EGNN) approach to achieve better few-shot learning performance with direct exploitation of both

intra-cluster similarity and the inter-cluster dissimilarity. In contrast, our approach achieves better performance without (a) needing additional graph neural network parameters and hence is more resistant to over-fitting, (b) requiring transductive setting, where collective test set is provided during inference which is anyway orthogonal to our contribution; see Sec 4 for comparative results.

## 4 EXPERIMENTS

**Overview.** In our experiments, we cover two main results: with and without distractors (see Sec 2.1), where distractors are present at train *and* test time when applied. In each, we discuss experiments with and without *semi-supervised adaptation* where additional unlabelled data are used at *test* time. Note that whether or not unlabelled data is available at test time, we use the same trained model, the difference comes from adding the adaptation step in Eq. 1 at test time to leverage that data.

### 4.1 EXPERIMENTAL SETUP

**Datasets.** We evaluated our work on the two commonly used SS-FSL benchmarks Omniglot and Mini-ImageNet. Omniglot (Lake et al., 2011) is a dataset of 1,623 handwritten characters from 50 alphabets. Each character was drawn by 20 human subjects. We follow the few-shot setting proposed by (Vinyals et al., 2016), in which the images are resized to $28 \times 28$ px and rotations in multiples of $90°$ are applied, yielding 6,492 classes in total. These are split into 4,112 training classes, 688 validation classes, and 1,692 testing classes. Mini-ImageNet (Vinyals et al., 2016) is a modified version of the ILSVRC-12 dataset (Russakovsky et al., 2015), in which 600 images, of size $84 \times 84$ px, for each of 100 classes were randomly chosen to be part of the dataset. We rely on the class split used by Ravi & Larochelle (2017a). These splits use 64 classes for training, 16 for validation, and 20 for test. In our experiments, following (Ren et al., 2018; Zhang et al., 2018), we sample 10% and 40% of the points in each class to form the labeled split for Omniglot and Mini-Imagenet, respectively; the rest forms the unlabeled split.

**Implementation Details.** We have provided full details of our experimental setting including network architectures, hyperparameter tuning on the validation set in appendixC. For fair comparison, we opt for the same Conv-4 architecture (Vinyals et al., 2016) appeared in the prior SS-FSL art (Zhang et al., 2018; Ren et al., 2018).

**Episode Composition.** All testing is performed on 5-way episodes for both datasets. Unless stated otherwise, the analysis performed in sections 4.2 & 4.3 are performed by averaging results over 300 5-shot 5-way mini-imagenet episodes from the *test* split, with $N_u$=10. Further detail is in appendix C. All accuracies reported are averaged over 3000 5-way episodes and reported with 95% confidence intervals.

**Baselines.** We evaluate our approach on standard SS-FSL benchmarks and compare to prior art; PN (Ren et al., 2018), MetaGAN (Zhang et al., 2018), and EGNN-Semi (Kim et al., 2019). We also compare PRWN with 3 control models; the vanilla prototypical network (PN) trained on the fully labelled dataset, denoted $PN_{all}$ (the oracle), which is considered to be our *target* model, a PN (Ren et al., 2018) model trained only on the labelled split of the data (40% of the labels), which is essentially PRWN without our random walk loss, and finally a PN trained with the state-of-the-art VAT (Miyato et al., 2018) and entropy minimization as a strong baseline; we denote it as $PN_{VAT}$.

### 4.2 SEMI-SUPERVISED META-LEARNING WITHOUT DISTRACTORS

For experiments without semi-supervised adaptation, we observe from the third horizontal section of Table 3, that PRWN improves on the previous state-of-the-art MetaGAN (Zhang et al., 2018), and EGNN-Semi (Kim et al., 2019) on all experiments, with a significant improvement on 5-shot mini-imagenet. It is worth mentioning that our PRWN has less than half the trainable parameters of MetaGAN which empolys an additional larger generator.

Experiments with semi-supervised adaptation are presented in bottom section in Table 3. Note that PRWN already improves on prior art without the adaptation. With the added semi-supervised adaptation, PRWN improves significantly, and the gap widens. On the 5-shot mini-imagenet task, PRWN achieves a relative improvement of 8,17%, 4,86%, and 8,28% over the previous state-of-the-art, (Ren et al., 2018; Liu et al., 2019; Kim et al., 2019), respectively.

Table 1: Semi-Supervised Meta-Learning + Ablation Study

| Model | Omniglot 1-shot | Mini-Imagenet | |
|---|---|---|---|
| | | 1-shot | 5-shot |
| $PN_{all}$(Snell et al., 2017) | 98.8 | 49.4 | 68.2 |
| PN (Ren et al., 2018) | $94.62 \pm 0.09$ | $43.61 \pm 0.27$ | $59.08 \pm 0.22$ |
| MetaGAN(Zhang et al., 2018) | $97.58 \pm 0.07$ | $50.35 \pm 0.23$ | $64.43 \pm 0.27$ |
| EGNN-Semi (Kim et al., 2019) | N/A | N/A | $62.52 \pm$ N/A |
| $PN_{VAT}$ (Ours) | $97.14 \pm 0.16$ | $49.18 \pm 0.22$ | $66.94 \pm 0.20$ |
| PRWN (Ours) | $\mathbf{98.28 \pm 0.15}$ | $\mathbf{50.89 \pm 0.22}$ | $\mathbf{67.82 \pm 0.19}$ |
| PN + Semi-supervised inference(Ren et al., 2018) | $97.45 \pm 0.05$ | $49.98 \pm 0.34$ | $63.77 \pm 0.20$ |
| PN + Soft K-means(Ren et al., 2018) | $97.25 \pm 0.10$ | $50.09 \pm 0.45$ | $64.59 \pm 0.28$ |
| PN + Soft K-means + cluster(Ren et al., 2018) | $97.68 \pm 0.07$ | $49.03 \pm 0.24$ | $63.08 \pm 0.18$ |
| PN + Masked soft K-means(Ren et al., 2018) | $97.52 \pm 0.07$ | $50.41 \pm 0.24$ | $64.39 \pm 0.24$ |
| TPN-Semi (Liu et al., 2018) | N/A | $52.78 \pm 0.27$ | $66.42 \pm 0.21$ |
| EGNN-Semi(T) (Kim et al., 2019) | N/A | N/A | $64.32 \pm$ N/A |
| PRWN + Semi-supervised inference (Ours) | $\mathbf{99.23 \pm 0.08}$ | $\mathbf{56.65 \pm 0.24}$ | $\mathbf{69.65 \pm 0.20}$ |

ANALYSIS

**Ablation study.** From Table 3, we can see that our PRW loss improves the baseline PN significantly, boosting the accuracy of PRWN up to 67.82% from 59.08% on 5-shot mini-imagenet for example. Moreover, while $PN_{VAT}$ proves a powerful model, competing with prior state-of-the-art, PRWN still beats it on all tests. Furthermore, We trained PRWN on mini-imagenet with only 20% of the labels, and we obtain an accuracy of 64.8% on the 5-shot task; outperforming the SOTA of 64.43% which uses double the amount of labels. Most remarkably, PRWN performs competitively with the fully labelled $PN_{all}$, even outperforming it on 1-shot mini-imagenet.

**Local & Global consistency Analysis.** To evaluate the global consistency, we take a look into the behavior of our random walker for our various models. We compute the landing probability over the graphs they generate: the probability a random walker returns to the starting prototype, given by $Trace(T^{(\tau)})$ from Eq. 2. We can see in Fig. 2a that even as $\tau$ grows, PRWN generates graphs with the highest landing probs. Following is $PN_{VAT}$, implying that enforcing local consistency still helps with global consistency. We can also see that $PN_{all}$ also does better than PN, indicating that the addition of extra labeled data also improves global consistency. To evaluate local consistency and adversarial robustness of our various models, we compute their average VAT loss. Unsurprisingly, $PN_{VAT}$ performs best with 1.1 loss, following are both PRWN and $PN_{all}$ with 3.1 & 2.91 respectively, then PN with 5.9. We see again that improving global consistency helps with local consistency, and so does additional labelled data.

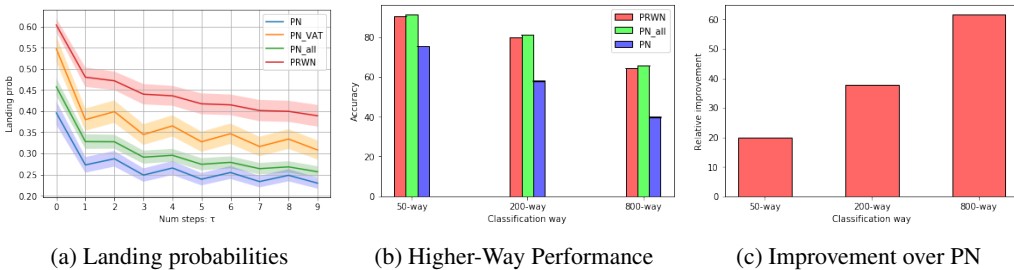

(a) Landing probabilities  (b) Higher-Way Performance  (c) Improvement over PN

Figure 2: *(a)* Landing Probabilities on mini-ImageNet: The $x$-axis denotes the number of steps for the walk ($\tau$), and the $y$-axis shows the probability of returning to the right prototype. *(b):* The Higher-Way performance on Omniglot as we increase the number of test classes $N_c$. *(c):* The relative improvement of PRWN over PN as we increase the number of classes in Omniglot

Table 2: Experiments with distractor classes

| Model | Omniglot | Mini-Imagenet | |
|---|---|---|---|
| | 1-shot | 1-shot | 5-shot |
| PRWN (Ours) | $97.76 \pm 0.11$ | $50.96 \pm 0.23$ | $67.64 \pm 0.18$ |
| PN+ Semi-supervised inference (Ren et al., 2018) | $95.08 \pm 0.09$ | $47.42 \pm 0.33$ | $62.62 \pm 0.24$ |
| PN+ Soft K-means (Ren et al., 2018) | $95.01 \pm 0.09$ | $48.70 \pm 0.32$ | $63.55 \pm 0.28$ |
| PN+ Soft K-means + cluster (Ren et al., 2018) | $97.17 \pm 0.04$ | $48.86 \pm 0.32$ | $61.27 \pm 0.24$ |
| PN+ Masked soft K-means (Ren et al., 2018) | $97.30 \pm 0.30$ | $49.04 \pm 0.31$ | $62.96 \pm 0.14$ |
| TPN-Semi (Liu et al., 2018) | N/A | $50.43 \pm 0.84$ | $64.95 \pm 0.73$ |
| PRWN+ Semi-supervised inference (Ours) | $\mathbf{97.86 \pm 0.22}$ | $\mathbf{53.61 \pm 0.22}$ | $\mathbf{67.45 \pm 0.21}$ |
| PRWN+ Semi-supervised inference + filter (Ours) | $\mathbf{99.04 \pm 0.18}$ | $\mathbf{54.51 \pm 0.23}$ | $\mathbf{68.77 \pm 0.20}$ |

**Discriminative Power.** In order to study our approach and baselines in a more challenging setup, we evaluate their performance on a Higher-Way classification. Fig. 2b shows that our model still performs better than the baseline and close to $PN_{all}$ (the oracle). The accuracy of PRWN, $PN_{all}$, and PN, on 800-ways in Omniglot, are 64.43%, 65.57% and 39.84%, respectively. In Fig. 2c, we show the relative improvement over PN reaching $\approx 60\%$ improvement on 800-ways classification. Similar behavior has been reported for mini-imagenet (Appendix F). This shows the performance gain from our PRW loss is robust and reflects its discriminative power.

**Transductive/Semi-supervised adaptation approaches. (Liu et al., 2019; Ren et al., 2018; Douze et al., 2018)** Our approach is orthogonal and can be integrated with these methods. In fact, PRWN + semi-supervised inference is such an integration where K-means step is integrated from (Ren et al., 2018). Tables 3, and 2 show that our network, combined with the K-means step at test time, perform far better than the networks trained with those adaptation methods. This supports our hypothesis that semi-supervised adaptation like the K-means step fails to fully exploit the unlabeled data **during meta-training**.

### 4.3 SEMI-SUPERVISED META-LEARNING WITH DISTRACTORS

The introduction of distractors by Ren et al. (2018) was meant to make the whole setup more realistic and challenging. To recap, the distractors are unlabelled points added to your support set, but they do not belong to any of the classes in that set i.e. the classes you are currently classifying over. This "labelled/unlabelled class mismatch" was found by Oliver et al. (2018) to be quite a challenge for SSL methods, sometimes even making the use of unlabelled data harmful for the model. We present our results in table 2, where the top row is our model without test time adaptation, and we can see that it already beats the previous state-of-the-art below, which makes use of test time unlabelled data, even by a large margin in the 5-shot mini-imagenet with a relative improvement of 3,8%, and 6,1% on TPN-Semi (Liu et al., 2019), and PN+Soft K-Means (Ren et al., 2018), respectively. Moreover, it beats the MetaGAN (Zhang et al., 2018) model trained without distractors on all tasks, and in fact performs closely to our own PRWN trained without distractors (*cf.* Table 3).

When we add the semi-supervised adaptation step, with distractors present among unlabelled data at test time, we see that our model does not benefit well from that step, and in the case of the 5-shot mini-imagenet, the performance is slightly harmed. In the next subsection, we will explore why our model is robust to distractors during training, and how we can use the random walk dynamics to make the semi-supervised inference step useful when distractors are present.

### DISTRACTOR ANALYSIS

We hypothesize that the reason our PRWN is robust against distractors, is because our random walker learns to largely avoid distractor points, and as such they are not magnetized towards our class prototypes; if anything by learning to avoid them, the network is structuring the latent space such that points of each class are compact and well separated. This comes as a by-product of the "prototypical magnetization" property that our loss models.

To test this hypothesis, we take a PRWN model trained with distractors, we sample test episodes including distractors ($N_d = N_c = 5$), construct our similarity graph, and compute the probability that our random walker visits distractor versus non-distractor points. Concretely, we compute $P = \frac{1}{N_c} \sum_{i=0}^{N_c} \Gamma^{(\mathbf{P} \rightarrow x)}$, where the summation is over the columns, and an entry $P_i$ represents the probability of visiting point $i$. We split $P$ into $P_{clean}$ and $P_{dist}$, containing the entries for non-distractor and distractor points. respectively. Both probabilities $p_{clean}$ and $p_{dist}$ should sum up to one. Whereas our baseline PN gets $p_{clean} = 0.67$, and PN$_{all}$ gets $p_{clean} = 0.76$, our PRWN gets $p_{clean} = 0.81$. So we see our $\mathcal{L}_{RW}$ is not only an attractive force bringing points closer to prototypes, but it also has a repelling force driving irrelevant points away from prototypes. Note this is not only a feature of the network, it is a property of the loss function. For instance, the semi-supervised inference step (Ren et al., 2018) involves all points, distractor or not, equally in the prototype update, regardless of the geometry of the embeddings.

**Distractors at semi-supervised inference.** We also performed an experiment to further improve PRWN with the semi-supervised inference step. We exploit our random walk dynamics to order to filter out distractors. We compute the probability that a point is part of a successful walk; a walk which starts and ends at the same prototype. This is given by $S = \sum_{i=0}^{N_c} \Gamma^{(\mathbf{P} \rightarrow x)} \odot \Gamma^{(x \rightarrow \mathbf{P})}$, where $\odot$ is the Hadamard product, and the summation is over the columns of the resulting matrix. Then we simply discard the points that scored below the median. With this little step, we see our PRWN + semi-supervised inference, become more robust to test time distractors, with 99.04% accuracy on omniglot, and 54.51% & 68.77% on mini-imagenet 1&5-shot, respectively. This simple filtering step just improved on the distractor state-of-the-art as shown in Table 2 (last row).

**More Distractors during Training.** Back to training, so far we have only explored models trained with $N_d = N_c = 5$. In this case there are as many distractor points as there are clean points. We stress test PRWN by training it on mini-imagenet with $N_d > N_c = 5$, namely $N_d = \{10, 15\}$. For 5-shot performance, we get accuracies of 66.92% & 66.75%, respectively. For 1-shot performance, we get 50.44% & 48.27%. We see that the model still manages to improve significantly over the baseline PN (Ren et al., 2018), despite 75% of the unlabelled points being distractors. In fact, with $N_d$ set to 10, PRWN still beats MetaGAN, the previous state-of-the-art model, trained on clean data in both mini-imagenet tasks(*cf.* Table 3). This shows that PRWN is a meaningful step towards real world settings, where unlabelled points are not guaranteed to come from relevant classes.

## 5 Conclusion

SS-FSL is a relatively unexplored yet challenging and important task. In this paper, we introduced a state-of-the-art SS-FSL model, by introducing a semi-supervised meta-training loss, namely the Prototypical Random Walk, which enforces global consistency over the data manifold, and magnetizes points around their class prototypes. We show that our model outperforms prior art and rivals its fully labelled counterpart in a wide range of experiments and analysis. We contrast the effects and performance of global versus local consistency, by training a PN with VAT (Miyato et al., 2018) and comparing it with our model. While the local consistency loss has an improvement on the performance, we found out that our global consistency loss significantly improves the performance in SS-FSL. Finally, we show that our model is robust to distractor classes even when they constitute the majority of unlabelled data. We show how this is related to the dynamic of PRW. We even create a simple distractor filter, and show its efficiency in improving semi-supervised inference (Ren et al., 2018). Our experiments and results set the state-of-the-art on most benchmarks.

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

## A  APPENDIX

## B  EPISODE CONSTRUCTION

---

**Algorithm 1** Construct a semi-supervised episode $E$, optionally with distractors.
RANDOMSAMPLE(S, N) denotes a set of N elements chosen randomly from set $S$, without replacement. Items sampled from the labeled split are tuples $(x_i, y_i)$, while items sampled from the unlabeled split are simply $(x_i)$

---

**Require:** $N_c$ {The number of classes or *way*}
  $N_s$ {The number of examples per class or *shot*}
  $N_q$ {The number of query images per class}
  $N_u$ {The number of unlabeled examples per class in the support set}
  $N_d$ {The number of distractor classes per episode}
  $V \leftarrow$ RANDOMSAMPLE($\{1 \cdots K\}$ , $N_c$)
  **for** $k \in V$ **do**
    $\mathcal{S}_{k,L} \leftarrow$ RANDOMSAMPLE($\mathcal{D}_{k,L}$ , $N_s$)
    $\mathcal{S}_{k,U} \leftarrow$ RANDOMSAMPLE($\mathcal{D}_{k,U}$ , $N_u$)
    $\mathcal{Q}_k \;\;\leftarrow$ RANDOMSAMPLE($\mathcal{D}_{k,L}$ , $N_q$)
    $\mathcal{Q} \leftarrow \mathcal{Q} \cup \mathcal{Q}_k$
    $\mathcal{S} \leftarrow \mathcal{S} \cup (\mathcal{S}_{k,L} \cup \mathcal{S}_{k,U})$
  **end for**
  **if** DISTRACTOR **then**
    $L \leftarrow$ RANDOMSAMPLE($\{1 \cdots K\}/V$ , $N_d$)
    **for** $k \in L$ **do**
      $\mathcal{S} \leftarrow \mathcal{S} \cup$ RANDOMSAMPLE($\mathcal{D}_{k,U}$ , $N_u$)
    **end for**
  **end if**

---

## C  EXPERIMENTAL DETAILS

**Architecture** We use the architecture from Vinyals et al. (2016) composed of four convolutional blocks. Each block comprises a 64-filter 3  3 convolution, batch normalization layer, a ReLU non-linearity and a $2 \times 2$ max-pooling layer. No other architectures were considered

**Optimization** We use ADAM for all our experiments with $\beta_1 = 0.9$, $\beta_2 = 0.99$. The initial learning rate for Omniglot experiments 0.001, and for all mini-imagenet 0.00025. For Omniglot, we cut the learning rate in half every 2K episodes. For mini-imagenet we cut the rate in half every 12500 episodes. For omniglot, we train for 20K episodes, for mini-imagenet we train for 100K episodes. For the learning rate the range considered was from 0.0001 to 0.001. For the decay steps the range considered was 2k to 20k. The maximun training step considered was 120K.

**Random walk** For all experiments the number of steps for our random walk $\tau = 3$, with an eponential decay $\alpha$ of 0.7, except on omniglot eperiments without distractors, where $\alpha = 1$. The weight given to for the semi-supervised loss $\lambda$ is 1.5 & 2 respectively for omniglot without and with distractors. For mini-imagenet, $\lambda$ is 0.5 for all experiments. We validated over $\tau$ in the range from 0 to 5, $\alpha$ from 0.5 to 1, and $\lambda$ from 0.5 to 2.5.

### C.1  EPISODE COMPOSITION

**Omniglot** For training without distractors, we use $N_c = 20$ and $N_u = 10$. For training with distractors, we use $N_c = N_d = 5$ and $N_u = 10$. And $N_s = 1$ in all cases. For all experiments, $N_q = 5$. For testing with additional unlabelled data, we use $N_u = 5$ for fair comparison with Ren et al. (2018).

**Mini-imagenet** For training we use $N_s = 5$, $N_c = 5$, and $N_u = 10$ for all experiments. When training with distractors, $N_d = 5$. For all experiments, $N_q = 5$. For testing with additional unlabelled data, we use $N_u = 20$ for fair comparison with Ren et al. (2018).

Table 3: Semi-Supervised Meta-Learning

| Model | Tiered-Imagenet | |
|---|---|---|
| | 1-shot | 5-shot |
| PN(Ren et al., 2018) | 46.52± 0.52 | 66.15± 0.22 |
| PRWN(Ours) | **54.87 ± 0.46** | **70.52 ± 0.43** |
| PN + Semi-supervised inference(Ren et al., 2018) | 50.74 ± 0.75 | 69.37 ± 0.26 |
| PN + Soft K-means(Ren et al., 2018) | 51.52 ± 0.36 | 70.25.59 ± 0.31 |
| PN + Soft K-means + cluster(Ren et al., 2018) | 51.85 ± 0.24 | 69.42 ± 0.17 |
| PN + Masked soft K-means(Ren et al., 2018) | 52.39 ± 0.44 | 69.88 ± 0.20 |
| TPN-Semi (Liu et al., 2018) | 55.74 | 71.01 |
| PRWN + Semi-supervised inference (Ours) | **59.17 ± 0.41** | **71.06 ± 0.39** |

# D    TIEREDIMAGENET

# E    LATENT SPACE VISUALIZATION

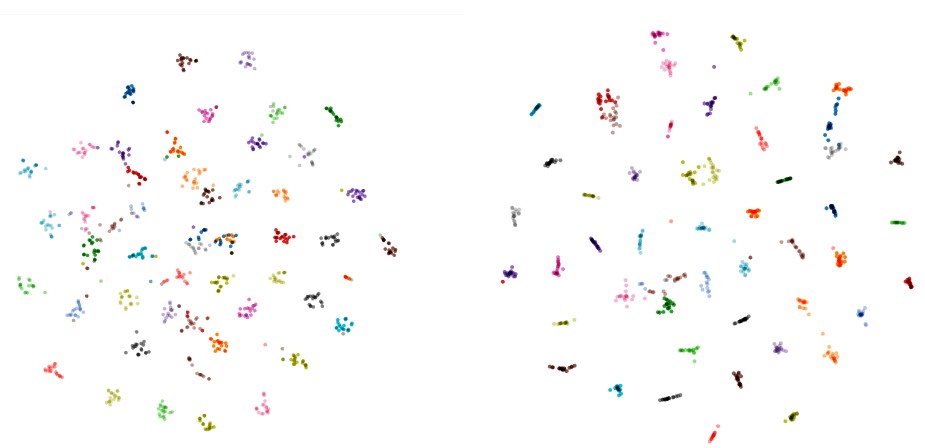

Figure 3: tSNE visualization of the embeddings for 50 classes on Omniglot. The embeddings of un-labeled data got magnetized to the class prototypes forming more compacted clusters in our PRWN *right* in contrast to the embeddings of PN *left*

# F    miniIMAGENET HIGHER WAY CLASSIFICATION

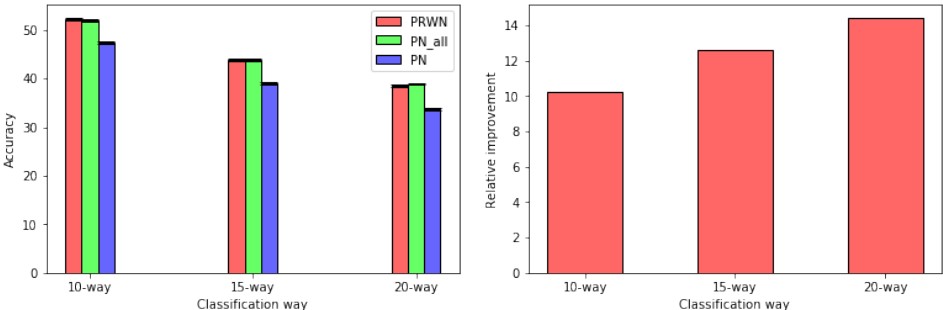

Figure 4: *Left:* Accuracy of our different models on 5-shot miniImageNet as we increase the number of classes $N_c$. *Right:* The relative improvement over PN as we increase the number of classes.

