# OpenReview forum: "Semi-Supervised Few-Shot Learning with Prototypical Random Walks"
_ICLR.cc/2020/Conference — Reject_

### Official Review · AnonReviewer2 · 2019-10-22
**Official Blind Review #2**

**Rating:** 3

**Review:**

This submission introduces a version of the random-walk regularizer introduced in Haeusser et al. 2017, in the setting of a Prototypical Network for semi-supervised few-shot learning. The authors show that using this regularizer, SOTA results can be obtained, notably on the popular miniImageNet. They also show that the random-walk can be used to successfully tackle the case with distractors.

I'm a bit on the fence for this submission. I could be convinced to accept, due to the impressive effectiveness of the method despite its simplicity. However, the submission is not without faults:

1. There are no results reported on tieredImageNet, despite it being used in the original paper on semi-supervised few-shot learning (Ren et al. 2018).

2. The proposed methods remains a fairly simple extension of the regularizer of Haeusser et al. 2017.

If the authors could address point 1 and add results on tieredImageNet, I would be willing to increase my rating.

Very small note: in the 4th section paragraph, is it possible that p(x_i|p_j) = \Gamma_{i,j}^{(p\rightarrow x)}, should be p(x_i|p_j) = \Gamma_{j,i}^{(p\rightarrow x)}?

**Experience Assessment:**

I have published in this field for several years.

**Review Assessment: Checking Correctness Of Derivations And Theory:**

N/A

**Review Assessment: Checking Correctness Of Experiments:**

I carefully checked the experiments.

**Review Assessment: Thoroughness In Paper Reading:**

I read the paper thoroughly.

---

> ### Author Response · Authors · 2019-11-15
> **Reply to R2**
>
> First, regarding results on Tiered-Imagenet, we have performed the experiments and updated the paper. A table is provided in the appendix with the results and will be later moved to the main paper. Unfortunately, we only
>  had computational time to perform the non-distractor, but we are currently running the distractor case and results will definitely be included in the final paper.
>  In the non-distractor case our results are SOTA, with 59% and 71% for the 1 and 5 shot respectively.
>
> We have addressed the novelty issue in the general comment.
>
> Regarding the final note, the reviewer is indeed correct, and we thank you for the note.

---

### Official Review · AnonReviewer1 · 2019-10-22
**Official Blind Review #1**

**Rating:** 3

**Review:**

This paper develops a random walk-based method on top of prototypical networks to address the semi-supervised few-shot learning, i.e. when each classification task can access only a few-shot labeled data but many unlabeled data. This paper defines a specific random walk: it walks from a prototype to an unlabeled sample, then walks for several steps between unlabeled samples, and then walk back to some prototype. They then propose two loss terms: one aims to maximize the probability of returning to the same prototype, and the other tries to make each unlabeled data having equal probability to be visited during the random walk. These two terms are added to the original prototypical network training loss and the resulted training procedure involves both the few-shot labeled data and the unlabeled data. Experiments on mini-ImageNet show that the proposed method outperforms the other baselines.

Intuitively, the main idea makes some sense to me. The writing is clear to understand the main idea but can be improved in various places. My major concerns are the motivation and analysis of the proposed random walk and the training algorithm, whose details cannot be found in the paper. The paper uses a lot of spaces to review the existing works and problem settings, but only spends one page (Section 2.2) on the description of their contribution, and the description lacks any in-depth discussion about “why it is designed in this way but not others” or “how this objective helps the few-shot learning”: it simply lists the procedures without convincing explanation.

Detailed comments:

1. It is called “random walks” but is different from the random walks used in classical methods in many key points and thus might not have all the existing properties of random walks. For example, it is not clear whether the negative of the square of Euclidean distance is a valid similarity for random walks. It is also not clear why the random walk probability matrix is computed by softmax instead of normalized Laplacian. Moreover, it fixes the starting nodes and end nodes, and also fixes the propagation steps: this cannot give any guarantee of any convergence (or asymptotic properties, mixing rate, etc) as the original random walks hold. This could be a serious problem because the random walks can stop at any non-stationary state after tau steps, which cannot be used for training.

2. It is not clear why the two additional loss terms (L_walker and L_visit) can help to improve the prototype training. The claimed motivation of minimizing these two terms cannot be soundly related to the final few-shot classification goal, though each of them independently makes some sense if applied to a different goal. Why does maximizing the probability of a prototype transit to itself give new information of improving few-shot classification? If prototypes are involved in the random walk nodes, why do we need to walk back to itself through many unlabeled nodes rather than simply walk to itself (which has the largest probability)? Why do we need to visit as many unlabeled data as possible? Isn’t the case that we hope to only visit the unlabeled data mostly related to the prototype and its class and rule out other unrelated ones since they introduce noises?

Here is a counterexample, which is a trivial solution that can minimize the two terms but does make any sense here: it is easy to define a model that produces the same prototype for different classes (so L_walker is minimized), and achieve equal distance between all the unlabeled samples (so L_visit is minimized), but this does not help the prototype training at all.

3. Another primary flaw here is that the training objective and the computational graph during training always change with the random walks, since the trajectory of the random walks keeps changing due to the randomness. Hence, the trained model does not have any guarantee and the convergence in the training is not the real convergence (it is not well-defined in the first place). If the random walks guarantee to converge to some stationary distribution, it might be possible to treat the varying computational graph as a stochastic one which has been analyzed in previous works. Unfortunately, due to the fixed number of steps and other issues mentioned above, this cannot hold. In Figure 2(a), we can see that even the largest tau they have tried cannot make the landing probability converge (not talking about the stationary distribution over all the nodes), which verifies my worries above.

4. Is the final performance sensitive to the hyperparameters lambda and tau? How did you choose their values in the experiments?

5. The methodology behind most existing semi-supervised few-shot learning methods, as also indicated in this paper’s introduction, is to apply or extend existing semi-supervised learning techniques on top of existing few-shot learning models. In fact, it is quite straightforward to do so and test all the semi-supervised learning techniques on few-shot learning tasks. Hence, I think it is necessary to compare to baselines that directly use, for example, consistency loss from mean teacher method and progressive training on pseudo labeling, or other SSL techniques mentioned in this paper: https://papers.nips.cc/paper/7585-realistic-evaluation-of-deep-semi-supervised-learning-algorithms.

-------------

Update:

Thanks for the authors' reply! After reading the rebuttal, I still think there are several main issues needed to be clarified or needing further study. Hence, I will keep my rating.

**Experience Assessment:**

I have published one or two papers in this area.

**Review Assessment: Checking Correctness Of Derivations And Theory:**

I carefully checked the derivations and theory.

**Review Assessment: Checking Correctness Of Experiments:**

I carefully checked the experiments.

**Review Assessment: Thoroughness In Paper Reading:**

I read the paper thoroughly.

---

> ### Author Response · Authors · 2019-11-15
> **Reply to R1**
>
> -We start with point 2. The motivation behind our l\_walker is to help train an embedding functions where "points of the same class form a compact cluster in latent space, well separated from other classes"(sec. 2.2). We trust it is clear why this helps with the final classification objective. We don't have labels to use in this loss, so the closest we can get is to enforce that the unlabeled points form compact well separated clusters around the class prototypes.
>
> The walker loss requires that any  walk starting at some prototype p, ends at the same prototype with probability one to reach its minimum. Let us denote this probability as the landing probability.
> To get close to the minimum, the graph must be structured such that if a point x has non trivial probability of being visited from a prototype p, then from x the probability of ending the walk in p should be close to 1. This means that if x is near p, it should be far away from all the other prototypes; each point should be close to a single prototype.
> This is not possible in the counter example provided in point 2, and in fact the walker loss would be quite high in this scenario. Since basically, which prototype we end the walk at would be independent from which prototype we started from; they are all the same. The landing probability would be 1/n for all prototypes in an episode with n prototypes.
>
> So the prototypes have to be far from each other. And from each prototype, we should only have non trivial probability of reaching points which lead back to that same prototype. Meaning each prototype, should have points surrounding it which are much closer to it, than to all the other prototypes.
>
> Which brings us to the visit loss. One potential problem with the walker loss, is that we could place most of the unlabeled points far from all prototypes, and place only two points close to each prototype. Then most of the random walks would be just between each prototype and its two neighbors. The visit loss is there to push against this tendency and include more points in the walks. It is important to note that the visit loss looks at the probability of points being visited from any prototype, not from each. Thus we do hope that to "visit the unlabeled data mostly related to the prototype and its class and rule out other unrelated ones since they introduce noises". Just that each point is visited from some prototype, this forces the model to simply not ignore the bulk of the unlabeled data.
>
> Thus to minimize both losses, we would have a graph where the prototypes are far from each other, and each prototype is surrounded by some of the points. Which is our stated goal.
>
> In the case where distractors are present, the visit loss does cause a certain tension, since we do not want to visit the distractor points from any prototype as they do introduce noise. This is a natural tension for SSL methods, as they do generally struggle with irrelevant unlabeled data.
>
> The best we can hope for here is that the two losses provide a balance, where relevant points are visited and irrelevant ones are still ignored.  Empirically, our model has excellent performance in the case of distractors. And we provide some analysis in sec 4.3 which shows how that the walker indeed avoids distractors.
>
> -We hope this clarifies the motivation of the graph based loss and our design choices. For instance, the reason we use a softmax over negative squared euclidean distances for our graph is because that is the similarity metric we are optimizing. This is the metric used by the PN to perform classification, and it is over this metric that we want to enforce our prior.
>
> -Regarding the convergence and random walk properties concerns mentioned in points 1 and 3, it is true that those are not random walks in the sense of methods such as spectral clustering or label propagation, however we are not using this walk for the same purpose or in the same setting. We have detailed our motivations above, and we do not assume convergence of the probabilities or any particular properties of random walks to hold to justify our loss. If the reviewer is still concerned with those issues after the clarifications, we would ask for a more concrete statement as to why the non-convergence of the probabilities or other lacking properties invalidate the training signal we derive here.
>
> -Regarding point 4, we have not conducted a full sensitivity analysis. We do We have not spent much time tuning those parameters, so it seems there is a wide range over which they give good performance. Also notable is that the hyperparameters work in a wide range of settings. For instance we have 8 experiments on mini-imagenet, all with SOTA results, and all with the same hyperparameters.
>
> Regarding point 5, we have already tested Virtual Adversarial Training(VAT), one of the currently most successful SSL techniques and show those results. However a full survey of SSL methods adapted to FSL is out of the scope of our paper.

---

### Official Review · AnonReviewer3 · 2019-10-23
**Official Blind Review #3**

**Rating:** 3

**Review:**

This paper suggests a method for semi-supervised few-shot learning. It is based on prototypical network, but in addition to the supervised loss a regularisation term that encourages unlabelled sample to be closer to the prototypes. This regularisation term is adapted from Haeusser et al. (2017) and is encouraging a random walk on a graph (where nodes are prototypes and unlabelled samples) that begins at a certain prototype to end at the same prototype.

The SS-FSL is indeed interesting and worth exploring. I also find the random walk on a graph of samples appealing and can potentially give insights on the structure of the embedding space. However, I think the paper is lacking in innovation, it is Haeusser et al. (2017) method applied to SS-FSL. The only difference is the usage of prototypes instead of labeled samples, this difference is also only relevant for 5-shot and not for 1-shot (where the prototype is the labeled sample) so there is only a single experiment (5-shot mini-imagenet) that is truly testing PRWN. Also, the choice of using prototypes (unlike Haeusser et al.) is neither justified theoretically nor empirically. Additionally, it is strange that the reported SS-FSL results are lower than standard FSL SOTA, e.g MetaOptNet (Lee et. al, 2019).


Update after rebuttal:
After reviewing the authors' response I'm changing my rating from reject to weak-reject


**Experience Assessment:**

I have published one or two papers in this area.

**Review Assessment: Checking Correctness Of Derivations And Theory:**

I assessed the sensibility of the derivations and theory.

**Review Assessment: Checking Correctness Of Experiments:**

I assessed the sensibility of the experiments.

**Review Assessment: Thoroughness In Paper Reading:**

I read the paper at least twice and used my best judgement in assessing the paper.

---

> ### Author Response · Authors · 2019-11-15
> **Reply to R3**
>
> We believe there may be a misunderstanding regarding the setting here from the ending statement, " it is strange that the reported SS-FSL results are lower than standard FSL SOTA, e.g MetaOptNet (Lee et. al, 2019".
>
> In SS-FSL settings the model gets strictly less training signal than in the FSL setting, as such it is natural that the performance is lower. Within the SS-FSL models published we have state of the art performance on all benchmarks, and the difference between our model and the runners up is often quite large as can be seen in the experiments tables. We have also investigated more challenging settings than previous works. For example, in the distractor analysis section we show experiments where 75\% of the points in the graph are distractors, irrelevant to the classes.
>
> Regarding the choice of having prototypes in the graph, the reasoning is quite simple. This graph optimizes exactly the properties we want our prototypical network to have: compact, well separated clusters around the prototypes. We have further detailed our intuitions in our comment for R1.

---

### Author Response · Authors · 2019-11-15
**General comment**

We would like to thank all the reviewers for their comments. Regarding the general concern regarding the novelty of the work. Our loss is in fact similar to Haeusser et al. (2017), but with some meaningful differences.

First, in Haeusser et al. (2017) the walker does not take any steps between the unlabeled points i.e. Tau is hardcoded to 0. In this respect, our loss is a generalization which accounts for higher order structure in the graph.

Second, our work is not simply introducing this loss, but in applying it in the FSL setting. This requires choosing the right methods from both fields, and integrating them well. For instance, TPN(Liu et al., 2018) is also a graph based SS-FSL method, however, in their work the authors chose to use the graph to propagate labels from the labelled point to the query points, then minimize the standard loss. Semi-supervised PN(Ren et al., 2018) use the same dynamic.
This basically amounts to using the unlabeled points in the adaptation stage. We hypothesized that this approach fails to fully exploit the unlabeled points for meta-training.
We on the other hand, derive a separate loss from the unlabeled points for meta-training. This is an important distinction and makes a large difference in performance as can be seen from comparing results. Our approach is can also be integrated with a semi-supervised adaptation step, as we have done by combining our approach with the kmeans step from (Ren et al., 2018). And we have shown that the model can benefit from both steps.

Finally, bringing this method to FSL allowed us to explore new venues. For instance, we show that this loss is robust against distractors.  We have shown SOTA performance with a model trained on graphs with 75% distractors.
We further explore the random walk dynamics in the presence of distractors in sec 4.3. We show that distractor points are less likely to be visited than clean points, we also show that we can use the dynamics of the random walker to effectively remove distractors from the episode with a simple heuristic. This is all quite novel.

In this light, we would hope that the reviewers would judge the novelty of the whole work, not just the loss itself.

---

### Decision · Program_Chairs · 2019-12-19

**Decision:**

Reject

**Comment:**

This paper proposed a semi-supervised few-shot learning method, on top of Prototypical Networks, wherein a regularization term that involves a random walk from a prototype to unlabeled samples and back to the same prototype.  SotA results were obtained in several experiments by using this method.  All reviewers agreed that the novelty of the paper is not such high compared with Haeusser et al. (2017) and the analysis and the experiments could be improved.